# The Agro-Pastoral Transitional Zone in Northern China: Continuously Intensifying Land Use Competition Leading to Imbalanced Spatial Matching of Ecological Elements

Kaige Wang [1,2], Xiangyu Zhao [1], Huihui Zheng [1], Bangyou Zheng [2], Yan Xu [1,3], Fengrong Zhang [1,3] and Zengqiang Duan [1,3,*]

1 College of Land Science and Technology, China Agricultural University, Beijing 100193, China; bs20223211063@cau.edu.cn (K.W.); zhxy@cau.edu.cn (X.Z.); zhenghuihui@cau.edu.cn (H.Z.); xyan@cau.edu.cn (Y.X.); frzhang@cau.edu.cn (F.Z.)
2 Agriculture and Food, Commonwealth Scientific and Industrial Research Organization, St Lucia, QLD 4067, Australia; bangyou.zheng@csiro.au
3 Key Laboratory of Agricultural Land Quality and Monitoring of Nature Resource, Beijing 100193, China
* Correspondence: duanzq@cau.edu.cn

**Abstract:** The agro-pastoral transitional zone (APTZ) in northern China is a typical ecologically vulnerable zone and a comprehensive geographical transitional zone. Its land use pattern has significant type diversity and spatial interlocking, which is always related to the play of ecological barrier functions and the sustainability of social-ecological systems. Accurately grasping the spatial competition relationship and comprehensive geographical effects of land types of APTZ in northern China is a key proposition for achieving coordinated and sustainable development between humans and land. To explore the spatial competition mechanism and comprehensive geographical effects of land use in the research area, this study focuses on revealing the evolutionary characteristics of land use patterns based on the center of gravity migration model. Based on the process of land use center of gravity migration, the spatial competition relationship of land types is explored to reveal the evolutionary trend and basic characteristics of land use in the APTZ. The results show the following: (1) Cultivated land and meadow are the main land types of the APTZ in northern China, accounting for up to 70% of the total regional area. The spatial competition between the two land types is the main contradiction in regional land use competition. (2) Drifting of the center of gravity of cultivated land towards the northwest direction is an important land use migration feature of the APTZ in northern China. Between 1980 and 2020, the center of gravity of cultivated land shifted by about 2 km to the northwest, and the center of gravity of grassland shifted by 8–10 km to the southwest. (3) The center of gravity of arable land and grassland in the entire region is constantly approaching, which has decreased from 70.95 km in 1980 to 61.38 km in 2020. The intensification of their interweaving has led to more intense land use competition. Grasping the basic characteristics and driving mechanisms of land type competition is an important means to achieve sustainable spatial governance. (4) The scale differentiation and regional differentiation characteristics of gradient effects are significant, and it is essential to prevent the risk of mismatch between land use and natural endowments in the northeast and north China sections. The research has surpassed the traditional method of analyzing land use competition, and by introducing a centroid model to analyze the spatial mechanism of land use competition, it has expanded the methodology for expanding research in the field of land science and provided basic references for regional sustainable development.

**Keywords:** agro-pastoral transitional zone (APTZ) in northern China; farmland; grassland; space competition; gradient effect; land use pattern

## 1. Introduction

The United Nations Sustainable Development Goals emphasize the coordination between economic growth and ecological protection [1]. Land use is an important element to regulate the relationship between economic growth and ecological protection. The fierce competition in land use often leads to ecosystem degradation and loss of human well-being, which is particularly evident in ecologically fragile zones [2]. Therefore, an orderly land use and management system is the foundation of sustainable development [3]. The agro-pastoral transitional zone (APTZ) in northern China is a comprehensive geographical transition zone and a typical ecologically fragile area, with land use exhibiting typical marginal and transitional characteristics [4,5]. Due to the weak stability, anti-interference ability, and restoration ability of the ecosystem in this region, the fragile natural background is prone to large-scale ecological degradation caused by human disturbance [6]. Therefore, the relationship between humans and nature in this region has long been in a state of sharp contradiction, which becomes a key focus of sustainable development research. The contradiction between human and nature is embodied in the spatial competition of land types [7,8]. As a natural cover in semi-arid areas, grassland can better adapt to the regional background environment. However, due to excessive human development of grasslands, the competition between cultivated land and meadow continues to manifest, leading to a series of land degradation problems, for instance, land desertification and reduction of biodiversity [9]. Therefore, accurately grasping the spatial competition relationship of land types and their comprehensive geographical effects in the APTZ in northern China is a crucial proposition for systematically identifying the essence of regional human–land relations.

This study of spatial competition features in land use belongs to the category of land use pattern research that aims to analyze the spatial structural representation and distribution of the methods, ownership, and degree of land use by human society at a certain scale [10–12]. This study of land use patterns has gradually evolved from the traditional study of land use spatial structure and function to the analysis of factor coupling processes and comprehensive effects [13,14]. On the one hand, research has expanded into more new fields and directions, with a greater emphasis on multi-factor coupling feedback, multi-process coupling mechanism elucidation, and exploration of multi-agent collaborative governance solutions [14,15]. On the other hand, traditional research on land use spatial structure and function also requires new methods and innovative systems [16]. This trend further requires that research on land use patterns should focus on the synthesis of scale effects, the construction of theoretical foundations, and the analysis of process coupling effects [17–19]. It is also more necessary to connect with sustainable governance practices [20]. This study of land use spatial competition is an extension and expansion of the connotation of land use pattern research [21,22]. Existing research is mostly based on econometric methods such as game theory models and opportunity constraint models to explore the degree of spatial competition and spatial correlation in land use, and gradually deepen study on the economic mechanism of land use competition and the game process of interest subjects [23,24]. The current focus of study on land use competition is on the benefits game of land use, exploring land use competition models based on economic frameworks. However, further exploration is needed on the characteristics of spatial distribution and mechanism effects of land use competition [25]. Existing research has explored a series of mature theories and methods around the evolution of land use patterns, which can better reflect the spatial structure evolution and functional transformation of land use [26–29]. The land use center of gravity model is a relatively mature method system that is widely used to characterize the overall migration trend of land use patterns [30]. However, further exploration is needed on how the center of gravity model relates to land use competition, which is also the key breakthrough issue of this study. There is an urgent need to construct a theoretical framework and innovate methodology in the study of land use competition mode characteristics and comprehensive geographical effects caused by competition [31–33]. This study innovates the methodological path in analyzing land use

patterns, which is closely related to the structural transformation, multifunctional evolution, and sustainability that mainstream global land science research focuses on.

The APTZ in northern China is a semi-agricultural and semi-pastoral area that transitions from semi-humid agricultural areas to semi-arid and arid pastoral areas [34,35]. Its land use has diversity in species, complexity in structure, spatial interlocking, and temporal alternation [36,37]. Previous studies have focused on analyzing regional land use competition from two perspectives: economic mechanisms and spatial patterns. In terms of economic mechanisms, the focus has been on the livelihood choices of farmers, the linkage of spatial behaviors among multiple entities, and the coordination of industrial policies [38]. The system has systematically revealed the driving mechanisms of economic factors behind land use competition in the APTZ [39,40]. In terms of spatial pattern paths, existing research has focused more on the traditional analysis of land use competition structure and spatial factor functions [41–43]. It is urgent to deepen the analysis of spatial mechanisms and comprehensive effects of comprehensive geographical elements that are highly transitional and critical in the APTZ, with the aim of revealing the spatial factor driving mechanism behind land use competition in the APTZ and serving regional sustainable development [44]. Research suggests that the drift pattern of the center of gravity for different land use types can reflect the interweaving and competitive correlation of spatial land class distribution. This study firstly analyzes the basic characteristics of the land use pattern in the APTZ in northern China, reveals the evolutionary characteristics of the land use pattern based on the center of gravity migration model, and then attempts to explore the spatial competition relationship of land types based on the process of land use center of gravity migration. The research objective of this study is to analyze the matching relationship between the characteristics of land use pattern changes and natural background conditions. This study also aims to reveal the evolutionary trend and basic characteristics of land use in the APTZ, and provide basic reference for scientific planning of the construction of ecological barriers in northern China and achieving regional sustainable development.

## 2. Materials and Methods

### 2.1. Overview of the Research Area

This study delineates the APTZ in northern China based on meteorological and administrative factors. Specifically, the northwest and southeast boundaries of the APTZ are defined as the range of annual precipitation between 250 and 450 mm, dryness index between 1 and 2, and precipitation variability between 15% and 30%. Considering the completeness of the statistical units for socio-economic data, some climate boundaries have been revised based on administrative boundaries. The east–west geographical span of the APTZ is relatively large, and the regional differences are significant. Therefore, this research area is divided into three zones.

The research area in this study covers an area of 793,800 square kilometers, spanning 9 provinces (cities or autonomous regions) (Figure 1). As typical land types in the APTZ, the proportion of cultivated land and grassland area is 20% and 50%, respectively (Table 1). The proportion of forest land is lower than that of cultivated land, and it is mainly distributed in the Greater Khingan Mountains in the northeast and the Yanshan Mountains in the north, without showing typical spatial interlocking. In terms of spatial distribution of land usage, the APTZ is bounded by the Great Khingan Mountains, the northern foothills of the Yanshan Mountains, and the southern edge of the Ordos Plateau. The northern side is a grassland concentration area, while the southern side is a cultivated grassland transitional distribution area. In terms of temporal evolution, from 1980 to 2020, the cultivated land area in this research area showed a trend of first increasing and then decreasing, while the grassland area showed a continuous decreasing trend. In summary, the northern APTZ has typical land type diversity and spatial transitional characteristics. Cultivated land and grassland are typical land types in this study area, and they are also the main transitional land types. Based on this, the spatial center of gravity model is used to further analyze the characteristics of land and grassland transitional and spatial competition.

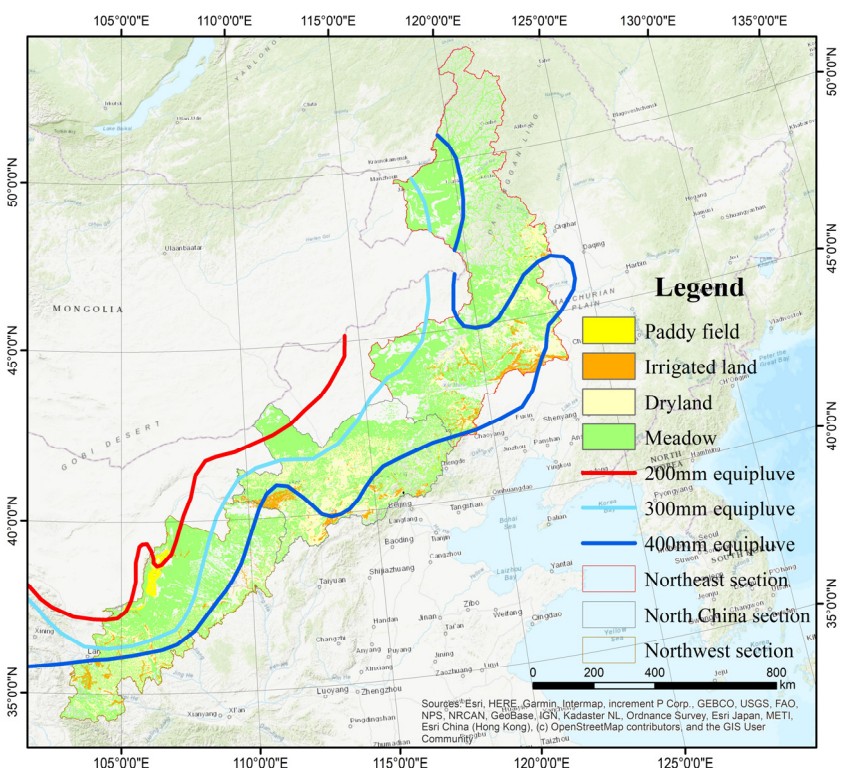

**Figure 1.** Position and main land use distribution of the APTZ in northern China.

**Table 1.** Changes in land use types in the APTZ from 1980 to 2020 (km², %).

| Age | Cultivated Land | Forest Land | Meadow | Water Area | Constructed Land | Unused Land |
|------|------|------|------|------|------|------|
| 1980 | 165,414 (20.84%) | 150,564 (18.97%) | 390,759 (49.23%) | 10,749 (1.35%) | 11,015 (1.39%) | 65,286 (8.22%) |
| 1990 | 168,081 (21.17%) | 149,506 (18.83%) | 388,333 (48.92%) | 10,439 (1.32%) | 11,585 (1.46%) | 65,843 (8.29%) |
| 2000 | 176,145 (22.19%) | 148,744 (18.74%) | 381,224 (48.03%) | 10,396 (1.31%) | 13,285 (1.67%) | 63,993 (8.06%) |
| 2010 | 173,752 (21.89%) | 151,274 (19.06%) | 379,754 (47.84%) | 10,058 (1.27%) | 14,627 (1.84%) | 64,322 (8.10%) |
| 2020 | 172,482 (21.73%) | 151,921 (19.14%) | 376,500 (47.43%) | 10,375 (1.31%) | 19,859 (2.50%) | 62,650 (7.89%) |

*2.2. Data Sources*

This study is based on a systematic analysis of multi-source data, which include five elements: climate, terrain, land use, administrative divisions, and policies (Table 2). The data are from the website of the Resource and Environment Data Center of the Chinese Academy of Sciences (https://www.resdc.cn/, accessed on 5 August 2023). This data platform publicly shares multiple data resources, with high spatial and temporal coverage and a complete range of data types, providing authoritative and accurate data sources for research in the fields of resource science and ecology. This study firstly delineates the boundaries of the APTZ in northern China based on climate and administrative data, then analyzes the features of land use change and its spatial center of gravity migration from 1980 to 2020, and finally combines climate, terrain, and policy data to analyze the reasons and gradient patterns for the forward shift of the land use center of gravity. This study integrates traditional data methods, extends beyond the basic idea of emphasizing structural and functional analysis in traditional land use research, and forms a new approach to land use pattern research by integrating the chain analysis method of "center of gravity analysis-competition mode analysis-gradient effect analysis". This composite method system further expands the traditional center of gravity model by analyzing the direction and distance of center of gravity migration to analyze the dynamic distribution of farmland and grassland, in order to clarify the matching relationship between land use change and natural local conditions.

| Name | Content | Accuracy | Usage |
|---|---|---|---|
| Climate data | China Land Annual Precipitation Frequency Distribution Parameter Dataset, China Meteorological Element Average Spatial Interpolation Dataset (annual precipitation and evaporation, annual average temperature), 1960–2020 | Spatial resolution of 1 km raster data | Defining the spatial scope of the APTZ and assisting in analyzing the spatial effects of land use center of gravity drift |
| Terrain data | Spatial distribution data of China's altitude (DEM), 2000 | Spatial resolution of 90 m raster data | Analyzing the spatial gradient effect of land use center of gravity migration |
| Land use data | Monitoring data of land use from remote sensing in China in 1980, 1990, 2000, 2010, and 2020 | Spatial resolution of 30 m raster data | Analyzing the land use center of gravity and its migration features |
| Administrative divisions data | Data from various provinces, cities, and county-level administrative regions in China, 2015 | Vector data | Defining the spatial scope of the APTZ and assisting in analyzing the reasons for the drift of land use center of gravity |

*2.3. Research and Methods*

2.3.1. Land Use Spatial Center of Gravity Measurement Method

The center of gravity has been drawn into the fields of geography and socio-economic research in recent years, and its connotation has been continuously spatialized to depict the distribution, location, and concentration characteristics of geographical and social factors in the spatial field. The evolution of land use spatial center of gravity can better reflect the spatial location migration characteristics and regional concentration of land use. This study analyzes the spatial location characteristics of land types in the northern APTZ by calculating the spatial center of gravity of different land use types, in order to support this study of spatial patterns of land use competition.

$$M_{(j,t)} = \sum_{u=1}^{n} \left[ GP_{(u,t)} * A_{(u,t)} \right] / \sum_{u=1}^{n} GP_{(u,t)} \tag{1}$$

$$N_{(j,t)} = \sum_{u=1}^{n} \left[ GP_{(u,t)} * B_{(u,t)} \right] / \sum_{u=1}^{n} GP_{(u,t)} \tag{2}$$

In the formula, $M_{(j,t)}$, $N_{(j,t)}$—the longitude and latitude coordinates of the gravity center for land use type j during period t; n—The total number of plots with land use type j in period t; $GP_{(u,t)}$—The area of the u-th plot with land use type j during period t; $A_{(u,t)}$, $B_{(u,t)}$—The geometric center longitude coordinates and latitude coordinates of the u-th plot of land use type j during period t.

The formula for gauging the distance of center of gravity drift in different periods of the same land use type is as follows:

$$D_{(u,m-n)} = \left\{ \left[ A_{(u,m)} - A_{(u,n)} \right]^2 + \left[ B_{(u,m)} - B_{(u,n)} \right]^2 \right\}^{1/2} \tag{3}$$

In the formula, $D_{(u,m-n)}$—The center of gravity movement distance of land use type $u$ during the m and n periods; $\left[ A_{(u,m)}, B_{(u,m)} \right]$, $\left[ A_{(u,n)}, B_{(u,n)} \right]$—The centroid coordinates of land use type u in the m and n periods.

### 2.3.2. Analysis of Competition Models for Land Use Spatial Center of Gravity

Core of gravity of land use reflects the spatial distribution centroid characteristics of land types, and its migration process marks the total migration features of land use. Different land use centers can show the interactive characteristics of land type distribution in the spatial field. The increase in spatial interlacing degree can enhance the spatial connectivity, patch complexity, and landscape heterogeneity between land types, which is an important incentive for triggering land use spatial competition (Tables 3 and 4). This study analyzes the spatial changes of land use competition based on the center of gravity drift pattern. Due to the physical characteristics of distance and direction, the center of gravity distance determines the intensity of land use spatial competition. The closer the centers of gravity are, the stronger the spatial adjacency and variability of land use, resulting in more intense land use spatial competition. The direction of gravity center drift characterizes the driving forces of spatial competition in different land types, which plays a crucial role in analyzing the driving factors of land type migration.

**Table 3.** Element characteristics of center of gravity drift.

| Centroid Element | Element Changes | Illustration | Meaning |
|---|---|---|---|
| Distance | Reduce |  | Distance between the centers of gravity decreases |
| | Expand |  | The geometric distance between the centers of gravity increases |
| Direction | Skew |  | Translate the two centers of gravity to the same starting point, with an angle of 0 to 180 degrees between the two motion axes |
| | Row-controlled |  | Translate the two centers of gravity to the same starting point, with the two axes of motion on the same direction axis. The Row controlled mode indicates that the center of gravity is subjected to a centripetal or centrifugal force. |
| | Coordination |  | The two axes of motion of the center of gravity are parallel, and the direction of motion is consistent |

Note: The red dots in the diagram represent the center of gravity of land type, and the arrows indicate the direction of migration.

**Table 4.** Joint analysis framework for "center of gravity drift ground competition".

| Distance | Direction | Land Competition Relationship |
|---|---|---|
| Approaching (reducing distance) | Skew Row-controlled Coordination | Eccentricity intensification Coaxial intensification Same direction intensification |
| Separation (distance expansion) | Skew Row-controlled Coordination | Heterogeneous slowing down Coaxial slowing down Same direction slowing down |

This study divides the Cartesian coordinate system into eight directional domains to clarify the directional characteristics of center of gravity transfer, with each domain spanning 45° (Figure 2). Based on the two basic elements of gravity center distance and direction, a "center of gravity spatial competition" analysis cascade framework is constructed. Finally, six land type competition relationships are formed based on the

combination of differentiated center of gravity distance and direction changes (Table 4). Combining the basic theories of physics and geography, the distance between the centers of gravity can reflect the degree of spatial proximity between two types of land. The proximity of the centers of gravity means that the degree of spatial overlap between the two types of land continues to increase, which characterizes the degree of competition between the two types of land. The direction of center of gravity migration marks the dominant direction of land use migration, which reflects the driving force direction of land use migration and can provide a detailed decomposition of the force direction for analyzing changes in land use patterns.

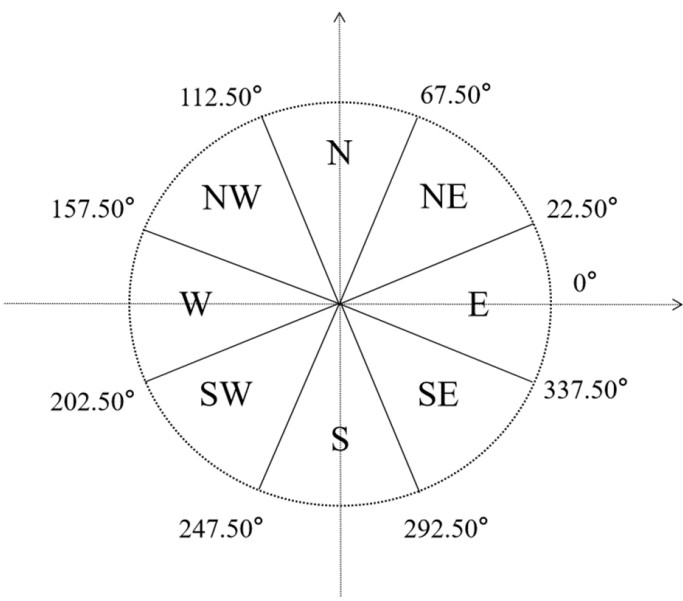

**Figure 2.** Definition of center of gravity drift direction domain.

2.3.3. Analysis of Gradient Effect of Land Use Center of Gravity Drift

The APTZ in northern China is a typical comprehensive geographical transition zone that exhibits a significant northeast–southwest trend [5]. Firstly, in terms of climate, both the 400 mm and 300 mm precipitation lines exhibit a "northeast–southwest" trend, exhibiting a typical gradient feature of regional precipitation gradually decreasing from the southeast to the northwest. Secondly, in terms of terrain, the APTZ in northern China is located in the transitional part between the second and third steps, which is in the core area of the "northeast–southwest" mountain range such as the "Greater Khingan Mountains- Yanshan Mountains- Yinshan Mountains- Helan Mountains" (Figure 1), presenting a gradient of gradual uplift from southeast to northwest. In terms of humanities and society, the famous "Hu Huanyong Line" runs through the agricultural and pastoral transitional zone in northern China from northeast to southwest, with a gradient feature of decreasing population density and transportation network density from southeast to northwest [5].

Based on the comprehensive gradient characteristics of the APTZ in northern China, the gradient effect caused by the drift of land use center of gravity can be determined. Due to the fact that the gradient surfaces of the comprehensive geographical elements all run in a northeast–southwest direction, the northeast–southwest direction domain (NE-SW) of the defined land use center of gravity migration direction domain is parallel gradient drift, while the other direction domains are cross gradient drift (Figure 3). Parallel gradient drift is the result of land use competition within the APTZ within the same gradient zone, while cross gradient drift indicates that cross gradient forces dominate land use change, and there may be mismatches and imbalances between land use and background conditions. The land use center of gravity drift model can provide methodological support for analyzing the

macro geographical effects caused by center of gravity drift, and provide basic guidelines for spatial planning and industrial structure adjustment.

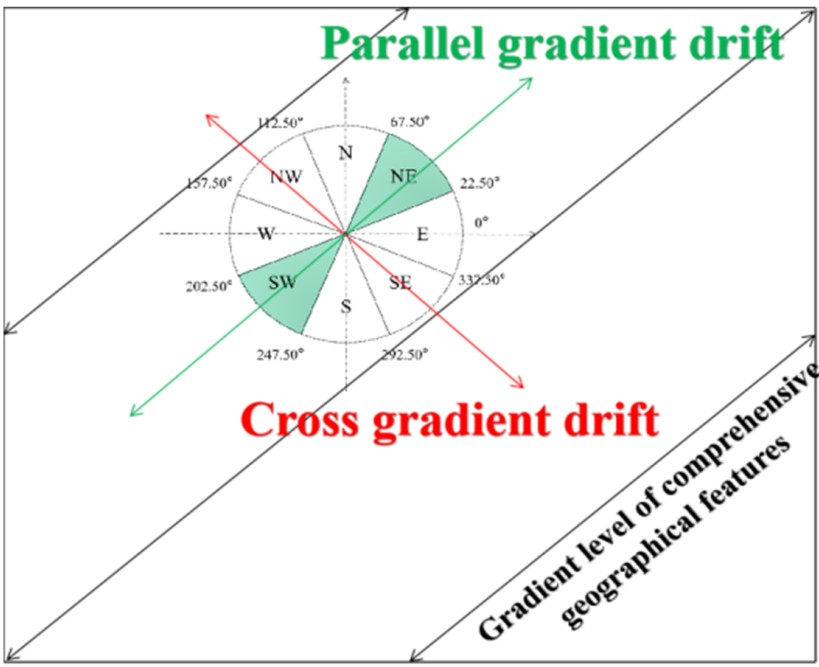

**Figure 3.** Mode analysis of gradient effect of center of gravity drift.

## 3. Results

### *3.1. Characteristics of Spatial Drift of Land Use Center of Gravity*

3.1.1. Spatial Drift Characteristics of Land Use Center of Gravity in the Whole Region

From 1980 to 2020, the center of gravity of farmland in the whole APTZ was distributed in Ulanqab City, central Inner Mongolia Autonomous Region (Figure 4a). Over the past forty years, the center of gravity of cultivated land has shown a trend of first drifting northeast and then southwest, with relatively small drift amplitudes within a distance of 2 km. Between 1980 and 2000, the center of gravity of cultivated land showed a significant shift towards the northeast direction, particularly between 1980 and 1990. From 2000 to 2020, the center of gravity of cultivated land showed a significant trend of drift towards the southwest, with an average drift distance of about 3 km. The overall focus of cultivated land shows a trend of migration towards the northwest direction.

From 1980 to 2020, the center of gravity of grassland in the whole APTZ was distributed in Shangdu County, central Inner Mongolia Autonomous Region (Figure 4b). Over the past forty years, the center of gravity of the grassland has shown a trend of drifting first to the southwest and then to the northeast, with a larger drift range than cultivated land, around 8–10 km. From 1980 to 2010, the center of gravity of the grassland continued to move significantly towards the southwest direction. Between 2010 and 2020, the center of gravity of the grassland shifted significantly towards the northeast direction. The overall center of gravity of the grassland showed a trend of migration towards the southwest direction. The center of gravity of the grassland is always on the north side of the center of gravity of the cultivated land.

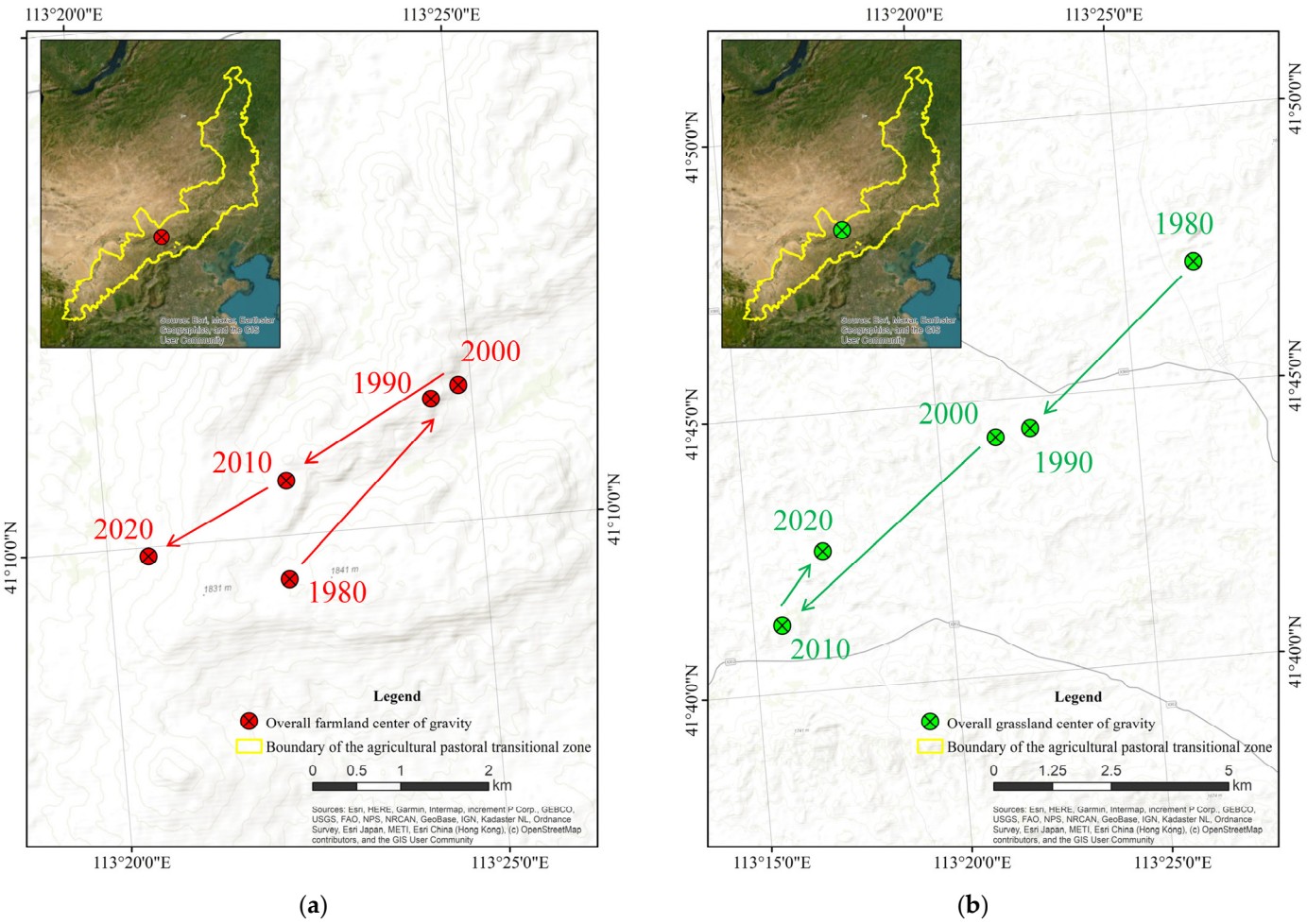

**Figure 4.** Distribution of land use center of gravity between farmland (**a**) and grassland (**b**) of the whole APTZ from 1980 to 2020.

3.1.2. Spatial Drift Characteristics of Regional Land Use Center of Gravity

Characteristics of the Northeast Section

In the northeast section of the APTZ, the center of gravity of cultivated land was located at the western foot of the Greater Khingan Mountains from 1980 to 2020, showing an overall drift trend from southeast to northwest (Figure 5a). The drift of the center of gravity of cultivated land towards the northwest direction was the most significant between 2000 and 2010, with a drift distance of 4 km. The process of drift towards the northwest also occurred between 1980 and 1990, as well as between 2010 and 2020.

In the northeast section of the APTZ, the center of gravity of the grassland was located in the western foothills of the Greater Khingan Mountains from 1980 to 2020, showing an overall drift trend from east to northwest (Figure 5b). The trend of continuous westward drift of the grassland center of gravity over the past 40 years is quite significant, with a drift distance of 7 km and a slowing trend in drift speed.

Characteristics of the North China Section

In the North China section of the APTZ, the center of gravity of cultivated land showed an overall drift trend from southeast to northwest from 1980 to 2020 (Figure 6a). The drift of the center of gravity of cultivated land towards the northeast direction was more significant between 1980 and 2000, with a drift distance of 5 km. After 2000, the drift towards the west direction was significant, and from 2010 to 2020, it was a typical drift towards the northwest direction.

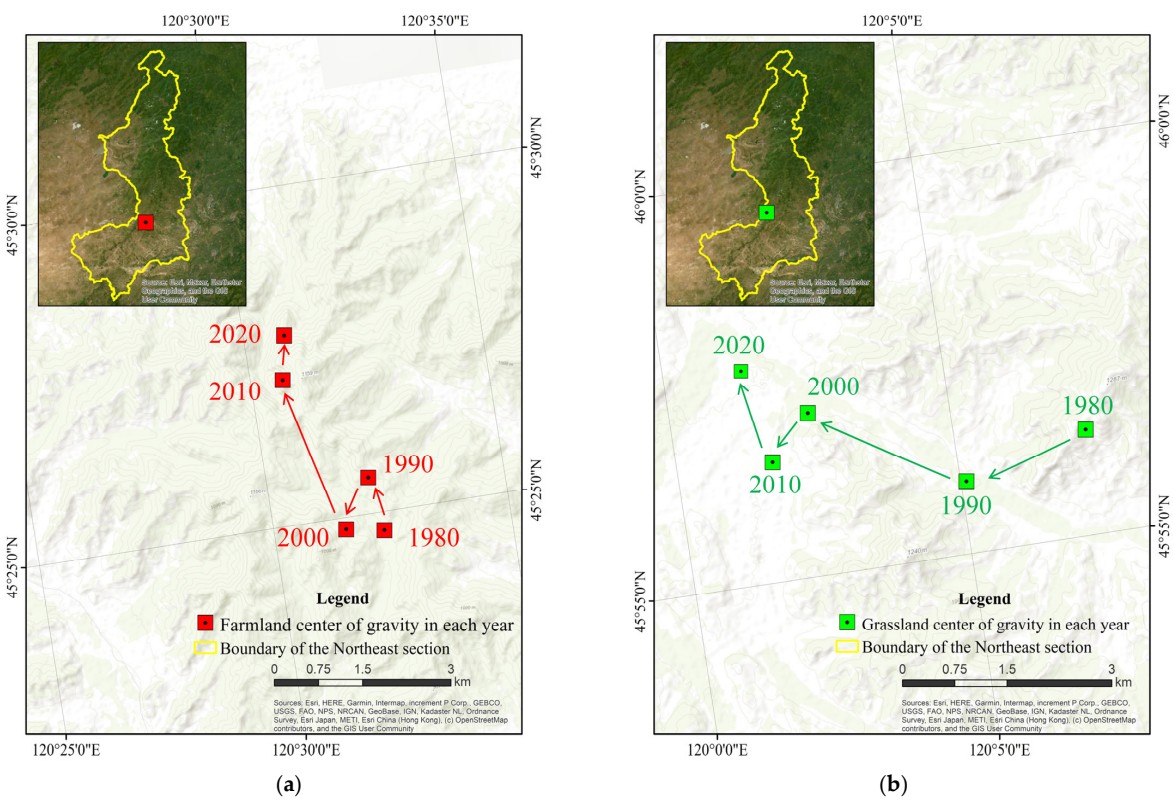

**Figure 5.** Spatial distribution of the center of gravity of cultivated land (**a**) and grassland (**b**) in the northeast section of the APTZ from 1980 to 2020.

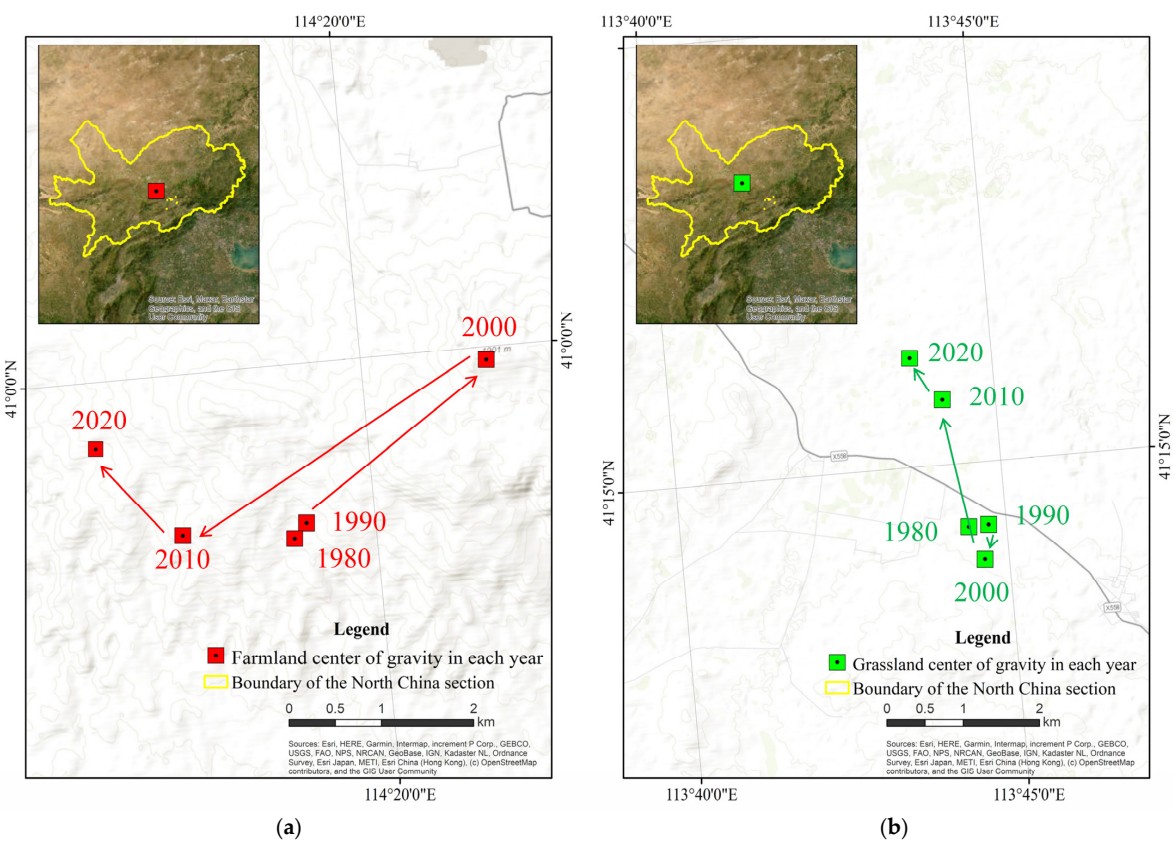

**Figure 6.** Spatial distribution of the center of gravity of cultivated land (**a**) and grassland (**b**) in the North China section of the APTZ from 1980 to 2020.

In the North China section of the APTZ, the center of gravity of the grassland showed an overall drift trend from southeast to northwest from 1980 to 2020 (Figure 6b). The trend of continuous westward drift of the grassland center of gravity over the past 40 years is quite significant, with a drift distance of 7 km and a slowing trend in drift speed.

Characteristics of the Northwest Section

In the northwest section of the APTZ, the center of gravity of cultivated land was located in the southern edge of transitional zone, showing an overall drift trend from northwest to southeast from 1980 to 2020 (Figure 7a). Among them, the drift of the center of gravity of cultivated land towards the northeast direction was more significant between 1990 and 2000, with a drift distance of 3 kms. After 2000, the drift direction turned south, and the drift process continued until 2020.

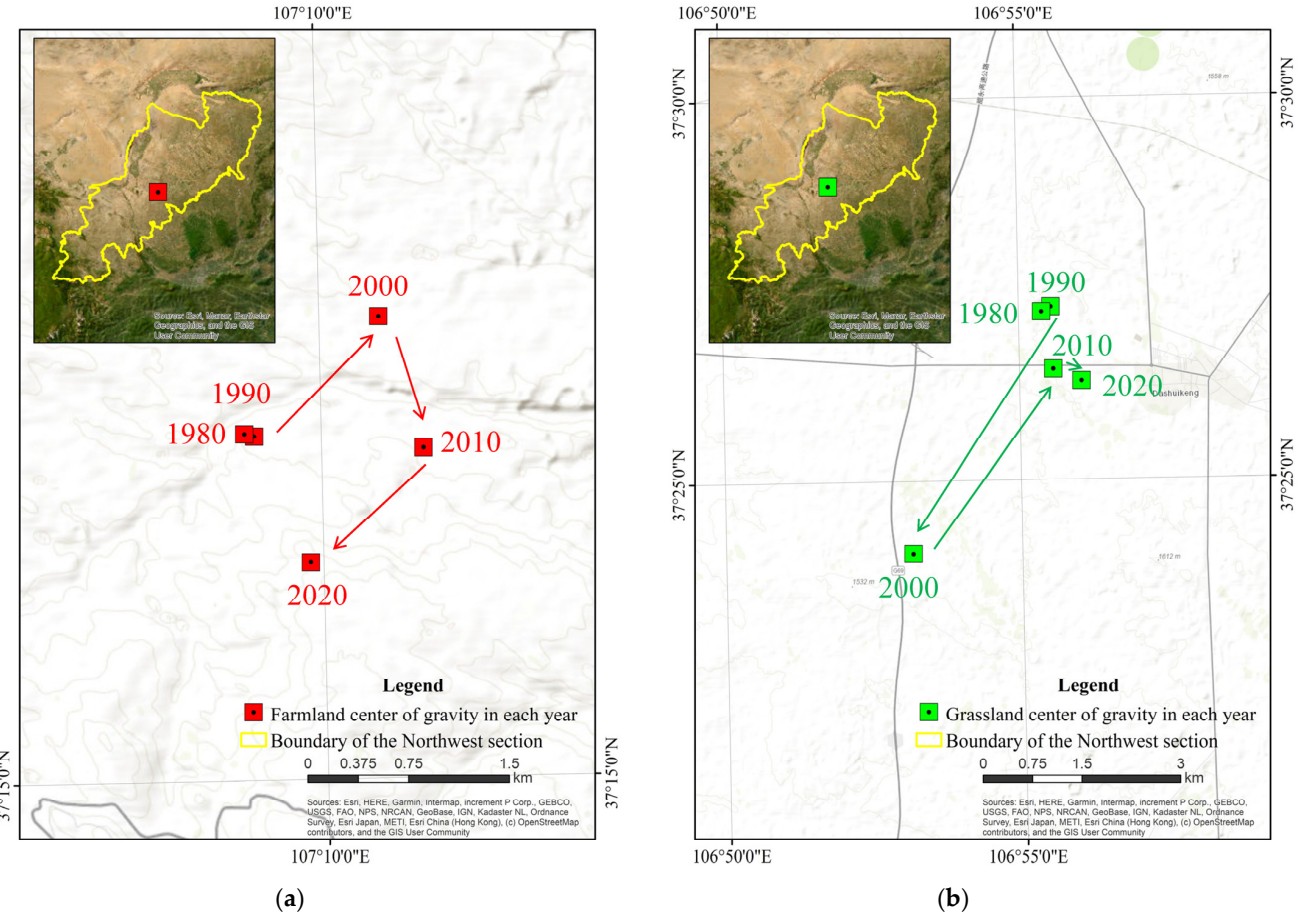

**Figure 7.** Spatial distribution of the center of gravity of cultivated land (**a**) and grassland (**b**) in the northwest section of the APTZ from 1980 to 2020.

In the northwest section of the APTZ, the center of gravity of the grassland was more northward compared to the center of gravity of cultivated land, and showed an overall drift trend towards the southeast from 1980 to 2020 (Figure 7b). Over the past 40 years, the center of gravity of the grassland experienced a significant drift towards the southwest around 2000, while in other years, the overall trend of drift towards the southeast was more significant.

### 3.2. Features of Spatial Competition in Land Use

#### 3.2.1. Spatial Competition Characteristics in the Whole APTZ

From 1980 to 2020, the center of gravity drift pattern in the whole APTZ showed a periodic fluctuation trend of "approaching and separating" in the distance dimension, but

still showed a drift trend of center of gravity approaching in the 40 year time series (Table 5). In the directional dimension, the center of gravity drift pattern is mainly focused on behavioral trends, indicating that the forces of land use change are on the same directional axis. The overall land use is significantly driven by coaxial forces, but the traction direction of the forces was opposite from 1980 to 1990, and from 2010 to 2020.

**Table 5.** Analysis of the center of gravity drift and spatial competition patterns in the whole APTZ.

| Year | Centroid Distance (km) | Direction of Grassland Gravity Center Movement | Direction of Cultivated Land Gravity Center Movement | Center of Gravity Drift Mode | Spatial Competition Mode |
|------|------------------------|-----------------------------------------------|-----------------------------------------------------|------------------------------|--------------------------|
| 1980 | 70.95 | ↙ | ↗ | Row-controlled–Approaching | Coaxial intensification |
| 1990 | 62.08 | ← | → | Row-controlled–Separation | Coaxial slowing down |
| 2000 | 62.09 | ↙ | ↙ | Coordination–Approaching | Exacerbation in the same direction |
| 2010 | 58.15 | ↗ | ↙ | Row-controlled–Separation | Coaxial slowing down |
| 2020 | 61.38 | | | | |

In the same period, there was a cyclical fluctuation in the competition mode of land use space, which intensified and slowed down, with the main direction of competition being coaxial competition. The overall trend of land use competition in the APTZ is intensifying, which to some extent indicates that there is a cyclical fluctuation in the short term and an increasing trend in the long term. Meanwhile, it is driven by coaxial forces and there is a certain degree of inter-regional sawing effect.

3.2.2. Spatial Competition Characteristics in Each Section
Characteristics of the Northeast Section

In the northeast section of the APTZ, the center of gravity drift pattern showed an overall trend of separation in the distance dimension from 1980 to 2020, which was particularly significant between 1980 and 2020 (Table 6). In the directional dimension, it exhibited a diagonal pattern, indicating that the changes in land use center of gravity in the northeast section were driven by opposite forces, forming a differentiated direction of center of gravity migration.

In terms of spatial competition patterns during the same period, the overall trend of land use spatial competition in the northeast section of the APTZ has slowed down. The intersection between farmland and grassland has shown a downward trend to a certain extent. In the direction of spatial competition, driven by opposite pulling forces, it can be clearly seen that land use competition is constantly migrating towards the "northwest–north" direction. This also shows that the development of cultivated land in the northeast section is continuously advancing towards the northwest direction, while the grassland as a whole is retreating towards the northwest direction.

**Table 6.** Analysis of the center of gravity drift and spatial competition mode in the northeast section of the APTZ.

| Year | Centroid Distance (km) | Direction of Grassland Gravity Center Movement | Direction of Cultivated Land Gravity Center Movement | Center of Gravity Drift Mode | Spatial Competition Mode |
|---|---|---|---|---|---|
| 1980 | 67.06 | ↙ | ↖ | Skew–Separation | Heterogeneous slowing down |
| 1990 | 68.07 | ↖ | ↙ | Skew–Separation | Heterogeneous slowing down |
| 2000 | 71.1 | ↙ | ↖ | Skew–Approaching | Eccentricity intensification |
| 2010 | 66.32 | ↖ | ↑ | Skew–Separation | Heterogeneous slowing down |
| 2020 | 67.33 | | | | |

Characteristics of the North China Section

In the North China section of the APTZ, the center of gravity drift pattern showed an overall approaching trend in the distance dimension from 1980 to 2020 (Table 7). This trend was particularly significant between 2000 and 2020, with a "coordination–skew" pattern in the directional dimension. In the periods of 1980–1990 and 2010–2020, land use competition was dominated by co-directional forces, while in the 1990–2010 period, it was dominated by opposite directional forces.

**Table 7.** Analysis of center of gravity drift and spatial competition patterns in the North China section of the APTZ.

| Year | Centroid Distance (km) | Direction of Grassland Gravity Center Movement | Direction of Cultivated Land Gravity Center Movement | Center of Gravity Drift Mode | Spatial Competition Mode |
|---|---|---|---|---|---|
| 1980 | 56.22 | ↗ | ↗ | Coordination–Approaching | Same direction intensification |
| 1990 | 55.69 | ↓ | ↗ | Skew–Separation | Heterogeneous slowing down |
| 2000 | 57.34 | ↖ | ↙ | Skew–Approaching | Eccentricity intensification |
| 2010 | 55.98 | ↖ | ↖ | Coordination–Approaching | Same direction intensification |
| 2020 | 55.10 | | | | |

In terms of spatial competition patterns during the same period, the overall trend of land use spatial competition in the North China section is intensifying, but there are differences in the direction of the driving force of competition in different periods. Between 1980 and 2000, the development of arable land in the eastern part of the North China region continued to increase, while the degradation of grasslands in the western part of the North China region led to a concentration of spatial competition between arable land and grasslands towards the northeast direction. In the past decade or more (2010–2020),

the development of cultivated land has been advancing to the grassland hinterland, and the area of dry farming on the Inner Mongolian Plateau has been expanding, resulting in the expansion of cultivated land to the northwest, the retreat of grassland to the northwest, and the concentration of land use competition to the northwest.

Characteristics of the Northwest Section

In the northwest section of the APTZ, the center of gravity drift pattern showed an overall trend of alternating "separation–approaching" in the distance dimension from 1980 to 2020, manifested as repeated tugging and sawing between land types, and the main pattern in the direction dimension was "skew". The force acting on the center of gravity drift was anisotropic (Table 8).

**Table 8.** Analysis of center of gravity drift and spatial competition mode in the northwest section of the APTZ.

| Year | Centroid Distance (km) | Direction of Grassland Gravity Center Movement | Direction of Cultivated Land Gravity Center Movement | Center of Gravity Drift Mode | Spatial Competition Mode |
|------|------|------|------|------|------|
| 1980 | 27.64 | | | | |
| | | ↗ | ↘ | Skew–Separation | Heterogeneous slowing down |
| 1990 | 27.66 | | | | |
| | | ↙ | ↗ | Row-controlled–Approaching | Coaxial intensification |
| 2000 | 27.63 | | | | |
| | | ↗ | ↘ | Skew–Separation | Heterogeneous slowing down |
| 2010 | 28.27 | | | | |
| | | ↘ | ↙ | Skew–Approaching | Same direction intensification |
| 2020 | 27.39 | | | | |

In terms of spatial competition patterns during the same period, the overall spatial competition between cultivated land and grassland in the northwest section is in a basic pattern of "slowing–intensifying" alternation, which is strongly related to the frequent changes in the direction of the forces it is subjected to. The large-scale policy of returning farmland to forests and grasslands has been implemented in the northwest section since 2000. Under this policy, grasslands have continuously expanded to the southeast in the past decade, while farmland has retreated to the southeast as a whole. Therefore, land use competition is concentrated in the southeast direction.

*3.3. Gradient Effect of Land Use Center of Gravity Migration*

3.3.1. Gradient Effect of Land Use Center of Gravity Migration in the Whole APTZ

From 1980 to 2020, the overall transfer of land use center of gravity in the whole APTZ showed a parallel gradient effect, with only a cross gradient effect observed between 1990 and 2000 (Table 9). This indicates that land use in the APTZ has always migrated in the "northeast–southwest" direction, and there has been no large-scale cross gradient land use migration. The changes in the land use pattern in the area are basically stable, and it is basically matched with the natural resource background and the gradient distribution pattern of human, social, and geographical elements.

**Table 9.** Gradient effect of land use center of gravity drift in the whole APTZ.

| Year | Direction of Grassland Gravity Center Movement | Direction of Cultivated Land Gravity Center Movement | Gradient Effect |
|---|---|---|---|
| 1980 | | | |
| | ↙ | ↗ | All are parallel gradient drift |
| 1990 | | | |
| | ← | → | All are cross gradient drift |
| 2000 | | | |
| | ↙ | ↙ | All are parallel gradient drift |
| 2010 | | | |
| | ↗ | ↙ | All are parallel gradient drift |
| 2020 | | | |

### 3.3.2. Gradient Effect of Land Use Center of Gravity Migration in Each Section of APTZ

From 1980 to 2020, the grassland land types in the northeast section of the northern APTZ showed an alternating trend of "parallel gradient effect–cross gradient effect", while the farmland as a whole showed a typical cross gradient effect (Table 10). The cross gradient effect of both land types showed significant drift towards the northwest direction, indicating that the center of gravity of farmland and grassland in the northeast section of the APTZ continued to drift towards the northwest. The utilization of arable land was gradually moving towards cooler areas and traditional pastoral areas, while grasslands were constantly retreating to the northwest. This cross gradient effect to some extent indicates that there is a spatial mismatch and imbalance between the land use pattern in the northeast section and the comprehensive natural, cultural, and geographical conditions. Arable land is expanding towards areas that are more unsuitable for agriculture, and grasslands are constantly being cultivated. Natural cover is largely disrupted by human cultivation.

**Table 10.** Gradient effect of land use center of gravity drift in the northeast section of the APTZ.

| Year | Direction of Grassland Gravity Center Movement | Direction of Cultivated Land Gravity Center Movement | Gradient Effect |
|---|---|---|---|
| 1980 | | | |
| | ↙ | ↖ | Parallel gradient drift of grassland center of gravity, cross gradient drift of farmland center of gravity |
| 1990 | | | |
| | ← | ↙ | Cross gradient drift of grassland center of gravity, parallel gradient drift of farmland center of gravity |
| 2000 | | | |
| | ↙ | ↖ | Parallel gradient drift of grassland center of gravity, cross gradient drift of farmland center of gravity |
| 2010 | | | |
| | ↖ | ↑ | Both land types exhibit cross gradient drift |
| 2020 | | | |

From 1980 to 2020, the land use in the North China section of the APTZ showed a strong parallel gradient effect in the early stage (1980–2000). The land use pattern remained basically stable. In the later stage (2000–2020), a typical cross gradient effect occurred, and the center of gravity of grassland and cultivated land showed a migration towards the northwest direction (Table 11). The North China section is the terrain transition area from the Inner Mongolian Plateau to the North China Plain where the Yanshan Mountains lie

across it. In the early stage, land development was basically carried out in the valley area at the north and south foot of the Yanshan Mountains. In the later stage, land development gradually expanded to the Inner Mongolian Plateau, causing the distribution of cultivated land to expand to the dry and cold areas in the northwest, and grassland to retreat to the northwest hinterland, which marks the spatial mismatch between land use change patterns and comprehensive natural, cultural, and geographical elements in the North China section.

**Table 11.** Gradient effect of land use center of gravity drift in the North China section of the APTZ.

| Year | Direction of Grassland Gravity Center Movement | Direction of Cultivated Land Gravity Center Movement | Gradient Effect |
|---|---|---|---|
| 1980 | ↗ | ↗ | Both land types exhibit parallel gradient drift |
| 1990 | ↓ | ↗ | Cross gradient drift of grassland center of gravity, parallel gradient drift of farmland center of gravity |
| 2000 | ↖ | ↙ | Cross gradient drift of grassland center of gravity, parallel gradient drift of farmland center of gravity |
| 2010 | ↖ | ↖ | Both land types exhibit cross gradient drift |
| 2020 | | | |

From 1980 to 2020, the overall grassland changes in the northwest section of the APTZ showed a parallel gradient effect, while the overall cultivated land changes showed a trend of alternating cross gradient and parallel gradient effects (Table 12). Unlike other regions, the northwest section has implemented the policy of returning farmland to green for a long time. Driven by this policy element, the center of gravity of farmland and grassland shows a cross gradient drift towards the southeast direction. The natural vegetation restoration of dryland in arid and semi-arid areas is a cross gradient effect that adapts to the natural resource endowment of the northwest section of the APTZ, which is "humid in the southeast and arid in the northwest". The land use pattern is more suitable for the spatial distribution of natural local conditions, and it also lays the foundation for strengthening the ecological barrier function of the northwest section of the APTZ.

**Table 12.** Gradient effect of land use center of gravity drift in the northwest section of the APTZ.

| Year | Direction of Grassland Gravity Center Movement | Direction of Cultivated Land Gravity Center Movement | Gradient Effect |
|---|---|---|---|
| 1980 | ↗ | ↘ | Parallel gradient drift of grassland center of gravity, cross gradient drift of farmland center of gravity |
| 1990 | ↙ | ↗ | Both land types exhibit parallel gradient drift |
| 2000 | ↗ | ↘ | Parallel gradient drift of grassland center of gravity, cross gradient drift of farmland center of gravity |
| 2010 | ↘ | ↙ | Cross gradient drift of grassland center of gravity, parallel gradient drift of farmland center of gravity |
| 2020 | | | |

## 4. Discussion

This study analyzed the spatial competition patterns and gradient effects of land use of the APTZ in northern China based on the drift of land use center of gravity. A methodological breakthrough has been achieved in the analysis of the spatial mechanism of land use competition and this study of spatial gradient effects. Firstly, traditional research on land use competition is based on economic mechanisms. This study analyzes the spatial mechanism of land usage competition using the spatial center of gravity model, and finds that the land use center of gravity can better reflect the interlocking nature of land types through the distance and direction factors of center of gravity migration. This method has good applicability in transitional zone areas with complex land use types. Secondly, this study analyzed the gradient effect of the center of gravity drift mode in the transition field of comprehensive geographical elements in the APTZ of northern China. It was found that the whole APTZ exhibited a typical parallel gradient effect, while the northeast and North China sections exhibited imbalanced cross gradient effects. The northwest section exhibited coordinated cross gradient effects driven by ecological policies. This study also found that the gradient effect, as a spatial distribution feature of geographical features, has a certain scale effect, which is clearly reflected in zoning studies. The APTZ did not show significant cross gradient effects throughout the region. At this macro scale, it mainly reflects the land type sawing effect between different regions within the transitional zone. The strong inter-regional sawing effect masks the cross gradient effect of land type changes within the region, thus demonstrating the scale of gradient effects. This study combines macro and meso scales using partitioning methods, respecting the differences in gradient effects at different scales.

It is worth noting that this study focused on analyzing the spatial competition relationship between farmland and grassland, and did not include forest land in the analysis system. There are three reasons for this: Forest land covers a relatively small proportion in the region and is not a typical land type; the distribution of forest land is relatively concentrated, mainly in the mountainous and hilly areas of the northeast section, and the spatial interlocking is not clearly reflected; and the utilization attributes of forest land are different from those of cultivated land and meadow, and are subject to more regulation by national engineering, regional planning, and forest and grassland policies.

The research results indicate that there is a mismatch between the migration direction of farmland in the current APTZ and the natural resource background conditions, which needs to be addressed by the management department. The continuous extension of arable land to more arid inland areas not only brings enormous water resource pressure, but also easily leads to land degradation. The contradiction between land use and natural background in terms of spatial effects still needs to be further studied in the future. The innovative application of the center of gravity model has made further attempts to answer this question.

This study has expanded the comprehensive methods for analysis of land use spatial patterns, providing strong methodological guidance for revealing the spatial effects and basic characteristics of land use in comprehensive geographic transition zones, which can be further refined in specific aspects. Firstly, the analysis and application of elements in the direction of center of gravity migration: the direction of land use center of gravity migration reflects the basic direction of spatial forces. In the future, more ecological and socio-economic factors can be coupled to analyze the dynamic mechanism of land use change, providing basic guidelines for rational planning of national land space. Secondly, research on gradient effects can further expand methods for evaluating land use suitability or sustainability. The APTZ in northern China has significant spatial element transition characteristics. Whether land use change adapts to this spatial element transition feature can be clarified by analyzing gradient effects. Future research on gradient effects can further strengthen quantitative and model automation identification and diagnosis. The innovation of this study mainly lies in expanding the methodological system of the center of gravity model, combining the forward shift characteristics of the center of gravity with

natural local conditions, paying attention to the distance and direction of the forward shift of the center of gravity, and analyzing the dynamic migration characteristics of land use from a more detailed decomposition process, providing a new methodological approach for the study of land use patterns.

## 5. Conclusions

This study integrates the analysis of land use center of gravity into the framework of land use competition research, and explores the spatial gradient effect of land use change in the APTZ of northern China, forming a cascading analysis framework with "center of gravity pattern competition mode spatial effect" as the core system, which provides a basis for regional planning and sustainable natural resource management and a new methodology for land use pattern research. The conclusions drawn from this study are as follows:

(1)  The drift of the center of gravity of cultivated land towards the northwest direction is an important land use migration feature of the APTZ in northern China. The changes in the land use center of gravity of cultivated land and grassland in the 40 year time series are different due to time and place, but the overall migration characteristics exhibited are that cultivated land continues to expand towards areas with more arid, cold, or poor precipitation stability in the northwest direction. The trend of cultivated land expansion squeezing the natural cover space of grassland still exists.

(2)  The intensification of the interweaving between cultivated land and grassland has led to more intense land use competition. The competition for land use in the whole APTZ has a certain periodicity, and the competition relationship between cultivated land and grassland in the North China section is constantly strengthening. The natural resource management department needs to pay attention to the disorderly expansion and extensive management of agricultural production in this region, do a good job in agricultural industry zoning and animal husbandry area planning, and achieve integrated and complementary development of agriculture and animal husbandry through specialized zoning and moderate mixed management models, avoiding the damage to natural ecology and the loss of human welfare caused by intensified land use competition.

(3)  It is necessary to prevent the risk of spatial mismatch between land use and natural endowments. Especially in the northeast and North China sections, the expansion of farmland towards the northwest direction is not suitable for the expansion of agriculture in arid or cold areas, and there is a spatial mismatch with the distribution gradient of geographical elements in the "northeast–southwest" direction of the APTZ. There is a greater possibility of the risk of imbalance between human activities and land in the northeast and north sections of the APTZ. Policymakers need to pay more attention to control the disorderly spread of arable land, and protect high-quality arable land while restoring natural grassland cover, which can promote coordination between nature conservation and human well-being.

The sustainable development of the APTZ requires a focus on the disorderly expansion of arable land while actively promoting the coupling of agriculture and animal husbandry in the industrial and value chains. Sustainable regional planning and policy control are effective guarantees that can provide a foundation for the synergy of regional industrial development and ecological protection.

**Author Contributions:** Conceptualization, K.W., Y.X., F.Z. and Z.D.; Methodology, K.W. and X.Z.; Software, K.W., X.Z. and H.Z.; Formal analysis, K.W., X.Z., H.Z. and B.Z.; Investigation, K.W. and Z.D.; Resources, K.W., X.Z., H.Z., B.Z. and Z.D.; Data curation, K.W., H.Z. and B.Z.; Writing—original draft, K.W. and Z.D.; Writing—review & editing, K.W., Y.X. and Z.D.; Visualization, K.W., Y.X. and Z.D.; Project administration, Y.X. and Z.D.; Funding acquisition, Y.X. and Z.D. All authors have read and agreed to the published version of the manuscript.

**Funding:** This research was funded by "Science and Technology Promoting Mongolia" Action Key Project, grant number [NMKJXM202303].

**Data Availability Statement:** Data are contained within the article.

**Conflicts of Interest:** The authors declare no conflict of interest.

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
