# Peer review of "The Agro-Pastoral Transitional Zone in Northern China: Continuously Intensifying Land Use Competition Leading to Imbalanced Spatial Matching of Ecological Elements"

_land, doi:10.3390/land13050654_

Round 1

Reviewer 1 Report

Comments and Suggestions for Authors

This study explores the spatial competition and geographical effects of land use patterns in the Agro-Pastoral Transitional Zone (APTZ) in northern China. Results reveal that cultivated land and meadow dominate the region, with drifting of gravity affecting land use and preventing mismatch between land use and natural endowments.

The relationship between economic growth and ecological protection is complex due to competition between land types. The agro-pastoral transitional zone (APTZ) in northern China is prone to large-scale ecological degradation due to human disturbance. Understanding the spatial competition relationship is crucial for sustainable development research.

This study of spatial competition features in land use is an extension of landuse pattern research, focusing on the spatial structural representation and distribution of land use by human society. It expands on traditional research, focusing on factor-coupling processes and comprehensive effects, and connecting with sustainable governance practices.

The APTZ in northern China transitions from semi-agricultural to semi-arid and arid coastal areas, with diverse landuse and complexity. Previous studies have focused on economic mechanisms and spatial patterns, but this study aims to deepen the analysis of spatial mechanisms and comprehensive effects for sustainable development.

THE METHODOLOGY IS NOT CLEAR 

From 1980 to 2020, the center of gravity of farmland utilization in northern Mongolia's Inner Mongolia Autonomous Region (APTZ) was in Ulanqab City. Over the past 40 years, cultivated land has drifted northeast and southwest, with small drift amplitudes.

From 1980 to 2020, the heart of gravity of cultivated land and grassland in the APTZ was located at the western foothills of the Greater Khingan Mountains, showing significant drift trends from east to northwest.

From 1980 to 2020, the core of gravity of cultivated land in the North China section of the APTZ showed a significant drift trend from southwest to northwest.

From 1980-2020, the heart of gravity of cultivated land and grassland showed a trend from northwest to southeast, with significant drift towards the southwest since 2000.

From 1980-2020, the global center of gravity drift pattern in the APTZ showed periodic fluctuations, with land use competition intensifying and cyclical, driven by coaxial forces and interregional awing effects.

Comments on the Quality of English Language

Minor editing of English language required

Author Response

This study explores the spatial competition and geographical effects of land use patterns in the Agro-Pastoral Transitional Zone (APTZ) in northern China. Results reveal that cultivated land and meadow dominate the region, with drifting of gravity affecting land use and preventing mismatch between land use and natural endowments.

#Reply  Experts have accurately summarized the research content of this article. Thank you.

The relationship between economic growth and ecological protection is complex due to competition between land types. The agro-pastoral transitional zone (APTZ) in northern China is prone to large-scale ecological degradation due to human disturbance. Understanding the spatial competition relationship is crucial for sustainable development research.

#Reply  Experts have accurately summarized the research significance of this article. Thank you.

This study of spatial competition features in land use is an extension of landuse pattern research, focusing on the spatial structural representation and distribution of land use by human society. It expands on traditional research, focusing on factor-coupling processes and comprehensive effects, and connecting with sustainable governance practices.

 #Reply  Experts have accurately summarized the research methods of this article. Thank you.

The APTZ in northern China transitions from semi-agricultural to semi-arid and arid coastal areas, with diverse landuse and complexity. Previous studies have focused on economic mechanisms and spatial patterns, but this study aims to deepen the analysis of spatial mechanisms and comprehensive effects for sustainable development.

  #Reply  Experts have accurately summarized the research objectives of this article. Thank you.

THE METHODOLOGY IS NOT CLEAR 

From 1980 to 2020, the center of gravity of farmland utilization in northern Mongolia's Inner Mongolia Autonomous Region (APTZ) was in Ulanqab City. Over the past 40 years, cultivated land has drifted northeast and southwest, with small drift amplitudes.

From 1980 to 2020, the heart of gravity of cultivated land and grassland in the APTZ was located at the western foothills of the Greater Khingan Mountains, showing significant drift trends from east to northwest.

From 1980 to 2020, the core of gravity of cultivated land in the North China section of the APTZ showed a significant drift trend from southwest to northwest.

From 1980-2020, the heart of gravity of cultivated land and grassland showed a trend from northwest to southeast, with significant drift towards the southwest since 2000.

From 1980-2020, the global center of gravity drift pattern in the APTZ showed periodic fluctuations, with land use competition intensifying and cyclical, driven by coaxial forces and interregional awing effects.

#Reply Methodology is the foundation of article analysis. The author has systematically organized and accurately expressed the methodology section based on expert opinions. The author focuses on a systematic introduction to the center of gravity model and its application methods. The corresponding details can be seen in the research methods section.

Reviewer 2 Report

Comments and Suggestions for Authors

This study integrates the analysis of land use center of gravity into the framework of land use competition research, and explores the spatial gradient effect of land use change in the northern APTZ of China. Overall, the paper is well structured, and the results are creative, providing a new methodology for land use pattern research. However, further modifications and improvements can be made in the following areas.

1.     Introduction. The introduction section contains too much content about the research area, which can be appropriately deleted to reduce redundancy.

2.     Methods. For the convenience of readers' understanding, you can further explain the six land type competition relationships in Table 4.

3.     Results. I suggest adding images to show the trend of land use spatial competition, making the results more intuitive.

4.     Discussion. The discussion section is the part that explains and analyzes the research results and findings. It is recommended that you provide more detailed supplementary explanations of the research results in the discussion section.

Comments on the Quality of English Language

The overall writing effect is good, but there are still some formatting issues. I suggest you make appropriate modifications to maintain consistency in the paper format.

Author Response

This study integrates the analysis of land use center of gravity into the framework of land use competition research, and explores the spatial gradient effect of land use change in the northern APTZ of China. Overall, the paper is well structured, and the results are creative, providing a new methodology for land use pattern research. However, further modifications and improvements can be made in the following areas.

  1. IntroductionThe introduction section contains too much content about the research area, which can be appropriately deleted to reduce redundancy.

#Reply  The author has made deletions to the introduction of the research area based on expert opinions. The detailed content can be obtained from the original text.

  1. For the convenience of readers' understanding, you can further explain the six land type competition relationships in Table 4.

#Reply Thank you for the expert's advice. The author has adjusted the expression of this method, emphasizing its basic characteristics and framework system, which can be viewed in the original text.

  1. I suggest adding images to show the trend of land use spatial competition, making the results more intuitive.

#Reply Thank you to the experts for providing valuable suggestions. Due to space limitations, increasing the number of images can lead to excessive text content. Therefore, the author combined expert opinions to optimize the existing images and systematically express them in the text introduction process.

  1. The discussion section is the part that explains and analyzes the research results and findings. It is recommended that you provide more detailed supplementary explanations of the research results in the discussion section.

#Reply Thank you to the experts for providing constructive feedback. The author has improved and perfected the discussion section, focusing on the scale effect of center of gravity transfer and sustainable development, in order to enhance the support for the development of discipline theory and practice.

Reviewer 3 Report

Comments and Suggestions for Authors

The paper is well written and it was very interesting to read the paper. Please consider the following comments to enhance the quality of the paper.

*The article lacks the objective of the study.

*In addition to the research objective, the paper should include a thesis/hypothesis along with questions to verify it. 

Please complete the research objective and research thesis/hypothesis along with the questions in the article.

This paper focuses on the processes occurring in land use in China. 

Please indicate (3-5 sentences )which elements of spatial matching of ecological in China are universal and which are attributed to local conditions land use in China?

Please add to the introduction what innovative research the authors propose? What has not yet been researched and what is the added value of this article in a new look at the development of y intensifying land use competition leading to imbalanced spatial matching of ecological element. What distinguishes yours research and gives it an innovative character?

*Please explain in the paper the extent to which the developed paper can be relevant to the international scientific field in particular  land use and ecological elements.

In your conclusions, please write

* What do the authors think should be the priority at the Agro-Pastoral Transitional Zone in Northern China?

*Add 3-4 sentences about sustainable development and the Agro-Pastoral Transitional Zone. 

Comments on the Quality of English Language

Requires a minor correction of some phrases.

Author Response

The paper is well written and it was very interesting to read the paper. Please consider the following comments to enhance the quality of the paper.

*The article lacks the objective of the study.

*In addition to the research objective, the paper should include a thesis/hypothesis along with questions to verify it. 

Please complete the research objective and research thesis/hypothesis along with the questions in the article.

#Reply Thank you to the experts for providing constructive feedback. The author has clearly defined the research objectives and emphasized in the article that the shift of focus between cultivated land and grassland signifies the overall trend of land use pattern changes. The research also strengthens the integration with landscape ecology and geography theories to enhance support for scientific issues and methodological frameworks.

This paper focuses on the processes occurring in land use in China. 

Please indicate (3-5 sentences ) which elements of spatial matching of ecological in China are universal and which are attributed to local conditions land use in China?

#Reply  This is a good advice. The ecological environment is directly related to the two major elements of climate and terrain, so climate and terrain elements can be used as basic indicators for analyzing land use and ecological environment conditions. Climate determines natural cover, while terrain reshapes it, resulting in several non zonal phenomena. This article focuses on exploring the coupling relationship between land use migration and the distribution of climate and terrain in China, in order to indicate the contradiction between land use migration in the current research area and the distribution of natural resources in China.

Please add to the introduction what innovative research the authors propose? What has not yet been researched and what is the added value of this article in a new look at the development of y intensifying land use competition leading to imbalanced spatial matching of ecological element. What distinguishes yours research and gives it an innovative character?

#Reply The current research on land use emphasizes the combination of sustainable development goals and regional background characteristics. This study combines land use and local characteristics of natural resources, and innovates the path of land use research by presenting land use competition through spatial means. The breakthrough of this article lies in placing the traditional focus model in the analysis of large-scale natural resource background, in order to clarify the relationship between land use migration and natural background, and then analyze land use sustainability. This is a new spatial perspective for analyzing sustainability.

*Please explain in the paper the extent to which the developed paper can be relevant to the international scientific field in particular  land use and ecological elements.

#Reply Thank you to the experts for their constructive suggestions. This study innovates the methodological path in analyzing land use patterns, which is closely related to the structural transformation, multifunctional evolution, and sustainability that mainstream land science research focuses on.

In your conclusions, please write

* What do the authors think should be the priority at the Agro-Pastoral Transitional Zone in Northern China?

#Reply This is a very constructive question. The research area should first strictly control the disorderly expansion of cultivated land. Secondly, we should pay attention to the coordinated development of agriculture and animal husbandry, and strengthen the exchange of material and information in the industrial chain. Finally, it is necessary to strengthen regional planning and control, and focus on balanced spatial development while considering the natural background. The above content has been presented in the original text according to expert opinions.

*Add 3-4 sentences about sustainable development and the Agro-Pastoral Transitional Zone.

#Reply Thank you to the expert for this valuable feedback .The author has presented it in the original text according to expert opinions. The author has optimized the conclusion and discussion section of the original text.

Round 2

Reviewer 1 Report

Comments and Suggestions for Authors

The revised edition of this manuscript showed no improvement. Furthermore, a discussion of the results compared to previous related works is still missing.

Author Response

Thank you to the expert for the valuable feedback. The author has optimized and adjusted the results analysis, discussion, and conclusion sections based on expert opinions.

In the results analysis section, the author conducted an accuracy check on the text expression, strictly following the order of whole APTZ , northeast section, North China section, and northwest section during the analysis, and corrected any inaccurate expressions in the text. In the result analysis, the image annotations in the text were also refined to achieve a close correspondence between the images and the text. In the discussion section, the author corrected and deleted inaccurate expressions, and added a connection between research and scientific issues.

In the discussion section, the author further enhances the coherence and clarity of the discussion, achieving a close connection with the research results.

In the conclusion section, the author further simplifies the content to ensure that each conclusion corresponds one-to-one with each result. The conclusion consists of three paragraphs. The first paragraph provides an overview of the status of land use center of gravity migration based on the results of section 3.1. The second paragraph clarifies the basic pattern of land use competition based on the results of section 3.2. The third paragraph indicates the mismatch between land use change and natural resource distribution based on the results of section 3.3.

The above revised content is reflected in the original text, please review.